TTC13 expression and STAT3 activation may form a positive feedback loop to promote ccRCC progression

Xie Lingling 1
Fang Yu 2
Chen Jianping 1
Meng Wei 3
Guan Yangbo guanyangbo123@163.com 3
Gong Wenliang gongwl_uro@163.com 2 4
1 Department of Laboratory Medicine, Affiliated Hospital of Nantong University , Nantong , China
2 Department of Urology, The First Affiliated Hospital of Naval Medical University (Shanghai Changhai Hospital) , Shanghai , China
3 Department of Urology, Affiliated Hospital of Nantong University , Nantong , China
4 Department of Urology, The Third Affiliated Hospital of Naval Medical University (Shanghai Eastern Hepatobiliary Surgery Hospital) , Shanghai , China
Uversky Vladimir
Electronic publication date: 2023 Oct 30
Publication date: 2023
Volume: 11
Electronic Location ID: e16316
Received 2023 May 25; Accepted 2023 Sep 28
Copyright: ©2023 Xie et al.
Copyright year: 2023
Copyright holder: Xie et al.
License: This is an open access article distributed under the terms of the Creative Commons Attribution License, which permits unrestricted use, distribution, reproduction and adaptation in any medium and for any purpose provided that it is properly attributed. For attribution, the original author(s), title, publication source (PeerJ) and either DOI or URL of the article must be cited.
License URL: https://creativecommons.org/licenses/by/4.0/

Keywords: TTC13, Clear cell renal cell carcinoma, Biomarker, Immunity, Prognosis, Cell autophagy

Funding: Nantong Commission of Health QN2022016 This study was funded by the Nantong Commission of Health (QN2022016). The funders had no role in study design, data collection and analysis, decision to publish, or preparation of the manuscript.

==============================
Background

Renal cell carcinoma (RCC) originates from renal tubular epithelial cells and is mainly classified into three histological types, including clear cell renal cell carcinoma (ccRCC) which accounts for about 75% of all kidney cancers and is characterized by its strong invasiveness and poor prognosis. Hence, it is imperative to understand the mechanisms underlying the occurrence and progression of ccRCC to identify effective biomarkers for the early diagnosis and the prognosis prediction.

Methods

The mRNA level of TTC13 was quantified by RT-PCR, while the protein level was determined by western blot and immunohistochemistry (IHC) staining. Cell proliferation was measured by cck-8, and cell apoptosis was detected by flow cytometry. The binding of STAT3 to the promoter region of TTC13 was determined by the luciferase reporter assay and chip experiments. STAT3 nuclear translocation was assessed by immunofluorescence staining.

Results

We found that TTC13 was up-regulated in ccRCC, and TTC13 promoted cell proliferation as well as inhibited cell apoptosis and autophagy of ccRCC through wnt/β-catenin and IL6-JAK-STAT3 signaling pathways. Furthermore, TTC13 might play a role in the immune infiltration and immunotherapy of ccRCC. Mechanistically, STAT3 activated the transcription of TTC13 gene.

Conclusions

STAT3 directly regulated TTC13 expression through a positive feedback loop mechanism to promote ccRCC cell proliferation as well as reduce cell apoptosis and autophagy. These findings suggested new and effective therapeutic targets for more accurate and personalized treatment strategies.

Introduction

Renal cell carcinoma (RCC) ranks as the seventh most frequently diagnosed cancer and the second most prevalent urinary system-related cancer worldwide. The incidence of RCC varies among different regions with the highest incidence in developed countries. RCC mainly includes three types, among which clear cell renal cell carcinoma (ccRCC) has the highest mortality rate (Piao et al., 2023). ccRCC is characterized by hematuria, pain, and lump in the kidney area mostly detected in its later stage, accounting for about 75% of all kidney cancers; nevertheless, ccRCC is often asymptomatic or insidious at its early stage. Although the diagnosis and the clinical treatment of ccRCC have been significantly improved during recent years, the prognosis of patients with advanced ccRCC is still suboptimal owing to the high risk of metastasis and poor response to radiotherapy and chemotherapy (Kase, George & Ramalingam, 2023). Therefore, understanding the mechanisms underlying the occurrence and progression of ccRCC is critical for the identification of biomarkers for early diagnosis, treatment selection, and prognosis prediction.

Tetratricopeptide repeat domain 13 (TTC13) is a member of the large tetratricopeptide repeats (TPR) family of proteins which consists of a degenerate, 34 amino acid repeats, and is expressed in 27 human tissues including the brain, bladder, heart, and lung. TPR-containing proteins are found not only in many organisms but also in various subcellular locations, such as cytoplasm, nucleus, and mitochondria. Functionally, the TPR domain plays a part in cell cycle, transcription and protein transport (Leontiou et al., 2019). Although the role of TPR-related proteins in tumors has been reported in leukemia, liver cancer, and gastric cancer (El-Daher et al., 2018; Shaheen et al., 2020), the function of TTC13 in tumors is not clear, and the expression and biological functions of TTC13 in ccRCC need to be determined.

In this study, we performed multiple bioinformatics analyses and validation experiments to explore the expression, biological functions, and prognostic value of TTC13 in ccRCC. We for the first time found that TTC13 was upregulated in ccRCC, and TTC13 expression was associated with several pathological features. Particularly, TTC13 could modulate immune infiltration and immunotherapy. Our findings suggested that TTC13 may act as a valuable independent predictive biomarker for the diagnosis of ccRCC. Our mechanistic studies indicated that TTC13 might contribute to ccRCC progression via regulating Wnt/β-catenin and IL6-JAK-STAT3 signal pathways. Taken together, TTC13 may play a critical role in ccRCC occurrence and progression, and TTC13 signaling axis may serve as new and effective therapeutic targets for the exploration of more accurate and personalized treatment strategies.

Material and Methods

Bioinformatics analysis

TTC13 expression data as well as the corresponding clinical data were obtained from The Cancer Genome Atlas (TCGA) database. The raw data were pre-processed using either log2 transformation or normalization and then were analyzed using R software v4.1.3. Differential gene expression of TTC13 was calculated using the “limma” R package (Wang et al., 2020), with a cut-off criterium of —log2 fold change(FC)—>1 and a false discovery rate (FDR) <0.05.

Human ccRCC tissue samples and cell lines

The tumor and adjacent non-cancerous tissues were obtained from the Affiliated Hospital of Nantong University. This study was approved by the hospital’s ethics committee (Institutional Review Board approval number: 2022-K003-02), and all patients offered written informed consents for the use of their samples. Normal HK-2 cell line and three ccRCC cell lines: A498, 786-0 and Caki-1 were obtained from either the Cell Bank of Chinese Academy of Sciences (Shanghai, China) or Procell Life Science & Technology Co. Ltd. (Wuhan, China). All cell lines were cultured according to the required culture conditions.

Antibodies

We used the following antibodies: TTC13 (AP13674a,abcepta,1:1000), Bax (ab32503, 1:1000; Abcam, Cambridge, UK), Bcl-2 (ab182858, 1:1000; Abcam, Cambridge, UK), IL-6(12912, 1:1000; Abcam, Cambridge, UK), cleaved-caspase-3 (9664, CST, 1:1000), LC3II/I (ab192890, 1:1000; Abcam, Cambridge, UK), P62 (88588, 1:1000; Cell Signaling Technology, Danvers, MA, USA), JAK2 (ab108596, 1:1000; Abcam, Cambridge, UK), Ki67 (ab92742, 1:1000; Abcam, Cambridge, UK), MMP9 (13667, 1:1000; Cell Signaling Technology, Danvers, MA, USA), phosphor-JAK2 (ab32101, 1:1000; Abcam, Cambridge, UK), phosphor-STAT3 (ab76315, 1:1000; Abcam, Cambridge, UK), STAT3 (ab68153, 1:1000; Abcam, Cambridge, UK), β-catenin (ab32572, abcam, 1:1000), GAPDH (5174, CST, 1:1000) and β-actin (ab8226, 1:1000; Abcam, Cambridge, UK).

Quantitative Real-Time Polymerase Chain Reaction (qRT-PCR)

qRT-PCR was carried out to examine the expression level of TTC13 in 64 paired ccRCC tumor and adjacent non-cancerous samples using ABI 7500. Experiments were performed in duplicate, and the thresholdcycle (CT) values were averaged. TTC13 gene expression was normalized to GAPDH expression resulting in the ΔCT value, where ΔCt = Ct Target − Ct GAPDH. The relative expression level was determined by 2−ΔCT as previously described (Aguilar-Briseno et al., 2020). The primers were synthesized by Tsingke Biotech (Shanghai, China), and the primer sequences were denoted as follows: for TTC13, forward 5′-GACTCAGACTGCGAACCCAA-3′ and reverse 5′- ACTTGGCCTGGCTCAGAATC-3′; for GAPDH, the forward primer sequence was 5′-GAGTCAACGGATTTGGTCGT-3′ and the reverse primer sequence was 5′-GACAAGCTTCCCGTTCTCAG-3′.

Cell transfection

shRNA for TTC13 gene silencing and the plasmid for TTC13 overexpression were purchased from GenePharma (Shanghai, China). Two shRNAs were used to exclude off-target effects: shRNA1 (5′-GCAGTGAATGACCTCACTAAA-3′) and shRNA2 (5′-GCTTACAGGAAGCCCTTAAGA-3′). The TTC13 shRNAs or TTC13 overexpression plasmid was transfected into ccRCC cells using the Lipofectamine 3000 reagent (L3000075, Invitrogen), and transfection efficiency was confirmed by western blot analysis. Cells were collected for in vitro functional experiments 48 h after transfection. Stable TTC13 knockdown of 786-0 cells was obtained by shRNA lentivirus infection and further puromycin selection (Tandon et al., 2018). shRNA for STAT3 gene silencing and the plasmid for STAT3 overexpression were also purchased from GenePharma (Shanghai, China).

Cell proliferation and apoptosis

Cell proliferation was determined by a CCK-8 detection kit (C0038; Beyotime, Haimen, China). Briefly, transfected cells were seeded into 96-well plate (5,000/well) and cultured for the indicated times. CCK-8 solution (10 µl) was added to each well at specific time points, and the absorbance at 450 nm was determined by a plate reader. Each experiment was independently repeated for three or five times. For apoptosis assay, the ccRCC cells were collected, stained with the annexin V-conjugated fluorescein isothiocyanate (FITC) and the propidium iodide (PI) (C1062S-2; Beyotime, Haimen, China) following the manufacturer’s instructions, and analyzed using FACScan™ flow cytometer (BD Biosciences).

Mouse tumor xenografts

Six-week-old male null mice (weighted about 18g) were obtained from Jihui Laboratory Animal Care Co., Ltd. (Shanghai, China) and were randomly separated into two groups for subcutaneous injection with 786-0 cells (2 × 106/200 µl PBS) either stably expressing NC empty vectoror TTC13 shRNA. The xenograft tumor growth was monitored every 5 days, and at day 35, the tumors were dissected, weighed, and subjected to immunohistochemistry (IHC) staining. All experimental procedures were performed in accordance with the institutional guidelines approved by the Shanghai Changhai Hospital, Naval Medical University.

Western blot analysis

Total proteins were extracted from ccRCC cells using the protein extraction kit (P0013B; Beyotime, Haimen, China) and quantified by the NanoPhotometer (Implen, Inc., Westlake Village, CA, USA). Then, the proteins were separated by the SDS-PAGE and transferred onto the PVDF membranes, followed by incubation with the corresponding primary antibodies. After extensive washing, the PVDF membranes were incubated with the secondary antibodies, and the specific protein bands were visualized by using an enhanced chemiluminescence (ECL) kit (P0018M; Beyotime, Haimen, China) and imaged in a gel imaging system.

Immunohistochemistry (IHC) staining

The protein expression and distribution of β-catenin, Ki67, MMP9, phospho-STAT3, cleaved caspase-3 and TTC13 were examined by IHC of paraffin-embedded sections of ccRCC and adjacent non-cancerous tissues. Briefly, after deparaffinization and rehydration, the paraffin slides were treated with citric acid buffer solution (pH =6.0) at 121 °C for 15 min, followed by treatment with 3% hydrogen peroxide for 20 min. Slides were blocked with 1% BSA for 15 min and then incubated with primary antibody (1:50 dilution) at 4 °C overnight. After extensive washing, the slides were incubated with goat anti-rabbit IgG-HRP secondary antibody (1:100 dilution) at room temperature for 1 h and then counterstained with hematoxylin for 30 s. Lastly, the conventional dehydration was performed, and the slides were examined as well as imaged under a microscope (DM500; Leica).

Immunofluorescence (IF) staining

Standard IF staining procedure was employed. Briefly, cells were seeded onto coverslips, fixed with 4% paraformaldehyde, permeabilized with 0.1% Triton, and then incubated with the primary antibodies at 4 °C overnight followed by incubation with the secondary antibodies for 1 h. The cells were subsequently stained with 0.1 µg/ml DAPI for 1 min at room temperature in the dark. The coverslips were mounted, and the images were acquired using a fluorescence microscope.

Luciferase reporter assay and chromatin immunoprecipitation (ChIP) assay

For luciferase reporter assay, Caki-1 cells treated with AG490 were transiently co-transfected with pGL3-basic-TTC13 reporter plasmid and pRL-TK expression construct using Lipofectamine 3000 reagent according to the manufacturer’s instructions. At 48 h after transfection, the cells were harvested, and the luciferase activity was quantified using the Bright-Glo™ Luciferase assay kit (E1910; Promega, Madison, WI, USA), which was normalized to the Renilla luciferase activity. Each experiment was performed in triplicate. For the Chip assay, a standard Chip assay protocol was used for cells crosslinking, nuclear isolation, and chromatin fragmentation. The fragmented chromatin was incubated with anti-STAT3 antibody at 4 °C overnight, and the eluted chromatin was subjected to quantitative PCR analysis. IgG was used as a negative control.

Statistical analysis

Statistical analyses were conducted using GraphPad Prism 8.0 and SPSS Statistics 22.0. Data were presented as median and standard error of the mean (SEM). To compare the overall survival between two groups, Kaplan–Meier (K-M) curves and the log-rank test were employed. Paired cases were carried out by using t-test, while the prognostic value of TTC13 was evaluated using the univariate and multivariate Cox regression analyses. A P-value <0.05 was regarded as statistically significant.

Results

TTC13 was upregulated in ccRCC

As shown in Fig. 1A, the expression level of TTC13 was different between various tumors and normal tissues with a significant upregulation in ccRCC (Fig. 1B) (P < 0.001), which was supported by the data from the paired tumor and non-cancerous samples (Fig. 1C). Consistently, qRT-PCR analysis (tumor =64, normal =64) showed that the mRNA level of TTC13 was significantly higher in ccRCC tissues than in normal tissues (P < 0.001) (Fig. 1D) which was further confirmed by western blotting (Figs. 1E and 1G) and IHC (Fig. 1F). Furthermore, the ROC curves were used to evaluate the efficacy of TTC13 expression for ccRCC diagnostic prediction, suggesting that TTC13 level could serve as a diagnostic biomarker (Fig. 1H). Moreover, the ccRCC patients were divided into low- and high-expression groups based on the median expression value of TTC13, and the K–M survival curve analysis showed that the overall survival of the high-expression group was worse than that in the low-expression group (P < 0.001) (Fig. 1I), indicating the negative association between TTC13 level and patient overall survival.

Figure 1 (A–I) Expression and clinical significance of TTC13 in ccRCC.

TTC13 promoted the proliferation and inhibited the apoptosis and autophagy of ccRCC cells

We next explored the biological functions of TTC13 in ccRCC. CCK-8 assay revealed that knockdown of TTC13 inhibited the proliferation of HK-2,786-0 and Caki-1 cells (Fig. 2A). On the other hand, the flow cytometry analysis revealed that knockdown of TTC13 increased the apoptosis of HK-2 and ccRCC cells (Fig. 2B). In support with these findings, western blotting results suggested that the levels of p62 and BCL-2 protein in shTTC13 transfected HK-2, 786-0 and Caki-1 cells were significantly decreased, while the levels of Bax and cleaved caspase-3 as well as the ratio of LC3-II/I was significantly increased. Conversely, TTC13 overexpression resulted in the opposite effects (Fig. 2C), suggesting that TTC13 was involved in the regulation of renal cancer cell survival and autophagy.

Figure 2 (A–C) TTC13 promoted the proliferation as well as inhibited the apoptosis and autophagy.

TTC13 silencing inhibited tumor growth in vivo

To explore the effect of TTC13 on tumor growth in vivo, we subcutaneously injected 786-0 cells transfected with either TTC13 NC or shRNA plasmid into nude mice and found that the tumor volume and weight of the shRNA group were evidently smaller than that of the NC group (Figs. 3A, 3B, 3C and 3D). Moreover, IHC staining showed that the expression of TTC13, Ki67, MMP9, β-catenin and p-STAT3 were lower while cleaved caspase 3 was higher in the shRNA group than in the NC group (Fig. 3E), which was further validated by western blot analysis (Fig. 3F). Together, these results indicated that knockdown of TTC13 effectively inhibited tumor growth in vivo.

Figure 3 (A–F) TTC13 silencing inhibited tumor growth in vivo.

Analysis of TTC13-related signaling pathways in ccRCC

To understand the role of TTC13 in the pathogenesis of ccRCC, GSVA analysis was performed by using TTC13-high or -low expression datasets to explore the TTC13-regulated signaling pathways. We identified TTC13 associated up- and down-regulated signaling pathways, including Wnt/β-catenin, IL6-JAK-STAT3, interferon-alpha, interferon-gamma, E2F targets signaling pathways (Fig. 4A), suggesting that TTC13 expression may be related to cellular immunity and cell cycle regulation. To validate the findings from bioinformatics analysis, we focused our in vitro experiments on two signaling pathways that have been well known to be involved in tumorigenesis. Specifically, we experimentally determined whether Wnt/β-catenin and IL6-JAK-STAT3 signal pathways were activated in ccRCC by examining the expression of TTC13, β-catenin, JAK2, p-STAT3, STAT3 and p-JAK2 in ccRCC tumor tissues as well as the adjacent normal tissues. Western blotting results demonstrated an activated Wnt/β-catenin and IL6-JAK-STAT3 signal pathways in ccRCC (Figs. 4B and 4C). To directly demonstrate the relationship between TTC13 and Wnt/β-catenin and IL6-JAK-STAT3 signal pathways, we assessed the effects of TTC13 overexpression or knockdown on these two signaling activities. We found that overexpression of TTC13 enhanced, while TTC13 knockdown inhibited, the expression of β-catenin, p-JAK-2 and p-STAT3 in ccRCC cells, suggesting the regulation of TTC13 in Wnt/β-catenin and IL6-JAK-STAT3 pathways (Fig. 4D).

Figure 4 (A–D) TTC13 related signaling pathways in ccRCC.

TTC13 contributed to ccRCC progression via Wnt/β-catenin and IL6-JAK-STAT3 signal pathways

Having demonstrated that TTC13 activated Wnt/β-catenin and IL6-JAK-STAT3 signal pathways in ccRCC, we speculated that TTC13 might contribute to ccRCC progression through the above two signaling pathways. To test this hypothesis, we performed rescued experiments using a specific inhibitor of Wnt/β-catenin ICG001 or IL6-JAK-STAT3 signaling pathway AG490 and found that the inhibitors could attenuate the growth-promoting effect of TTC13 (Figs. 5A and 5C). Western blot analysis showed the similar results (Fig. 5D). Furthermore, we determined the dose response curve of cells to a 3-day inhibitor treatment and revealed that TTC13 overexpression increased the IC50 of AG490 and ICG001, indicating that TTC13 enabled ccRCC cells to resistant to drug treatment (Fig. 5B). Taken together, the above results demonstrated that TTC13 promoted ccRCC progression at least partly through activating Wnt/β-catenin and IL6-JAK-STAT3 signal pathways.

Figure 5 (A–D) TTC13 promoted ccRCC growth through Wnt/ β-catenin and IL6-JAK-STAT3 signal pathway.

STAT3 activated the transcription of TTC13 gene

More importantly, we investigated the molecular mechanisms underlying the elevated TTC13 expression in ccRCC. Since we observed that TTC13 could activate the IL6-JAK-STAT3 signaling pathway, we postulated that STAT3 might enter the nucleus upon TTC13 overexpression. Indeed, immunofluorescence staining confirmed a significant translocation of STAT3 from cytoplasm to nucleus when TTC13 was overexpressed (Fig. 6D). Since STAT3 is a transcription factor, we next explored whether STAT3 affected the activity of the TTC13 promoter. In line with this notion, transcription factor binding profile database JASPAR has identified several potential STAT3 binding sites on TTC13 promoter (http://jaspar.genereg.net/) (Figs. 6A and 6B). Consistently, TTC13 expression was decreased by STAT3 knockdown while TTC13 expression was increased by STAT3 overexpression (Fig. 6C). To further demonstrate the transcriptional regulation of TTC13 by STAT3, we carried out the luciferase reporter assay and found that the TTC13 promoter-driven luciferase activity in AG490-treated Caki-1 cells were significantly reduced (Fig. 6F). Chip assay further confirmed that STAT3 directly bound to the TTC13 promoter (Fig. 6E), suggesting that STAT3 might directly regulated TTC13 expression through a positive feedback loop mechanism to promote ccRCC cell proliferation, as well as to reduce cell apoptosis and autophagy.

Figure 6 (A–F) STAT3 regulated TTC13 expression at the transcription level.

Discussion

ccRCC is the most prevalent histological subtype, accounting for more than 75% of all diagnosed kidney tumors, and has the characteristics of strong invasiveness and poor prognosis (Narisawa et al., 2023; Ye et al., 2023). Tetratricopeptide repeat domain 13 (TTC13) is a member of a large family of proteins named tetratricopeptide repeats (TPR), which contains more than 5,000 members. So far, to the best of our knowledge, there is no report on the expression and functions of TTC13 in ccRCC. In this study, we analyzed the expression level and the prognostic value of TTC13 in ccRCC as well as explored its biological functions via both the bioinformatics analysis (Figures 1s–5s) and the experimental confirmation. Our experimental results showed an upregulation of TTC13 at both mRNA and protein levels in ccRCC cells as well as in ccRCC tissues. In addition, ccRCC patients with high TTC13 expression had poor prognosis. Furthermore, overexpression of TTC13 promoted the proliferation of ccRCC cells, while inhibited the apoptosis and autophagy of cells. Hence, our results suggested that TTC13 might play a key role in the occurrence and progression of ccRCC (Fig. 7).

Figure 7 A working model of TTC13 regulation in ccRCC cells.

Drawing material from Figdraw (http://www.figdraw.com; this material ID: UOAUR7ce55).

Accumulating evidence has revealed that the dysregulation of the Wnt/β-catenin signal pathway contributes to the development and progression of several solid tumors and hematological malignancies (Di Bartolomeo et al., 2023; Han et al., 2023; Li et al., 2022; Muto et al., 2023). In this study, we discovered that the Wnt/β-catenin signal pathway was also abnormally expressed in ccRCC, suggesting the involvement of Wnt/β-catenin in ccRCC. In addition, the IL6-JAK-STAT3 pathway is abnormally overactivated in numerous cancer types, which is often associated with poor outcomes (Ni et al., 2020; Siersbaek et al., 2020; Zhan et al., 2021). In this study, we found that the IL6-JAK-STAT3 signal pathway was activated in ccRCC, suggesting the therapeutic significance of this pathway. In support with our findings, Zhan et al. (2021) also identified the IL6-JAK-STAT3 signal as a potential risk factor in ccRCC by univariate and multivariate Cox regression analysis.

Our subsequent study demonstrated that overexpression of TTC13 could activate the IL6-JAK-STAT3 and Wnt/β-catenin signal pathways, whereas knockdown of TTC13 suppressed these two signaling pathways. Further experiments revealed that TTC13 promoted ccRCC cell proliferation and restrained apoptosis or autophagy through IL6-JAK-STAT3 and Wnt/β-catenin signal pathways. Consistent with our findings, Wang et al. (2021a) and Wang et al. (2021b) had reported that CENPA promoted the progression of ccRCC by activating the Wnt/β-catenin signal pathway. In addition, a recent study had revealed the Wnt/β-catenin signal-induced ARL4C expression in ccRCC (Zhang et al., 2022). On the other hand, several studies had indicated that some genes acted as tumor suppressors by inhibiting the Wnt/β-catenin signal pathway in ccRCC. For example, Gorka et al. (2021) had reported that β-catenin in ccRCC cells was significantly reduced at both mRNA and protein levels by MCPIP1 overexpression. In line with these findings, Xu et al. (2022) demonstrated that the upregulation of SDHA resulted in a significant suppression of the Wnt/β-catenin signal pathway by decreasing the β-catenin expression in ccRCC. Moreover, the activator of Wnt/β-catenin signal pathway can attenuate the inhibition on the malignancy of ccRCC cells caused by TLN2 overexpression (Cai et al., 2022). And SOX17 displayed the similar function as TLN2 (Wang et al., 2021a). These results, together with our findings, provide supporting evidence that Wnt/β-catenin signaling pathway is a crucial regulator in the progression of ccRCC, highlighting the clinical significance of targeting this signaling pathway. As for IL6-JAK-STAT3 signaling pathway, consistent with our findings, Wang et al. (2018) also reported that IL-6 and p-STAT3 expressions in renal cell carcinoma tissues was obviously higher compared with adjacent normal tissues. Another study found that knockdown of circ_0000274 RNA expression significantly reduced the protein levels of p-JAK1/JAK1 and p-STAT3/STAT3 in 786-0 and A498 cells, while inhibiting miR-338-3p expression reversed this effect (Qi et al., 2022). In addition, the conditioned medium of TAMs increased the phosphorylation level of STAT3 in RCC cells (Chen et al., 2022). Furthermore, it has been reported that tumor-associated macrophages promote RCC epithelial-mesenchymal transition, migration, and invasion via activating IL-6/STAT3 signaling. Consistent with these findings, another study showed that the total pSTAT3 and nuclear pSTAT3 levels were prominently increased in ccRCC tissues compared with the adjacent tissues (Song et al., 2019). Chae et al. (2020) also reported that Thymoquinone effectively prevented the phosphorylated form of STAT3 from entering the nucleus and binding to DNA to activate the transcription of target genes. Similarly, SIRT1 destabilized STAT3 through the ubiquitin-proteasome pathway, resulting in a decreased STAT3-dependent FGB expression, which in turn inhibited RCC cell proliferation (Chen et al., 2019). Collectively, these findings indicated that TTC13 may be associated with suppressed antitumor immune responses in the tumor microenvironment of ccRCC. Therefore, therapies targeting TTC13 as well as IL6-JAK-STAT3 signaling pathway may benefit ccRCC patients by simultaneously suppressing tumor cell growth and stimulating anti-tumor immunity.

One important finding of our study was that STAT3 bound to the promoter of TTC13 gene to upregulate the expression of TTC13, which in turn further activated the JAK2/STAT3 signal pathway to increase the nuclear import of STAT3, thereby forming a positive feedback loop to promote the progression of ccRCC. A recent investigation had also revealed that the JAK/STAT3 signaling pathway regulated RCC cell apoptosis and glycolysis through RNF7, as STAT3 directly binded to RNF7 promoter (Xiao et al., 2022). Taken together, these data suggested that IL6-JAK-STAT3 signal pathway played a significant role in the pathogenesis of ccRCC, providing the rationale of targeting this pathway in ccRCC treatment.

Nonetheless, this study also had some limitations. First, we used retrospective data from public databases, which needs further validation in larger cohorts of ccRCC patients with well-defined clinical staging and sufficient clinical data. In addition, the biological function of TTC13 in ccRCC need to be further investigated. Lastly, it is necessary to improve and standardize the detection method of TTC13 gene to increase the feasibility of clinical application.

Conclusions

In conclusion, we were the first to use a variety of bioinformatics methods and verification experiments to explore the expression and clinical value of TTC13 in ccRCC. Our results indicated that TTC13 may play a role in the proliferation, apoptosis, and autophagy of ccRCC. In addition, TTC13 may serve as a novel biomarker for the diagnosis and prognosis prediction for patients with ccRCC.

Supplemental Information

Supplemental Information 1 Uncropped Blots

The original image of WB.

Click here for additional data file.

Supplemental Information 2 Bioinformatics related codes for TTC13 gene

Click here for additional data file.

Supplemental Information 3 Code for TTC13 gene

Click here for additional data file.

Figure 1S TTC13 expression is associated with 8 clinicopathological characteristics

Click here for additional data file.

Figure 2S GSEA diagram of TTC13 related signaling pathways

Click here for additional data file.

Figure 3S The associations between TTC13 and immune cell

Click here for additional data file.

Figure 4S TTC13 predicts the immune response and drug sensitivity

Click here for additional data file.

Figure 5S TTC13 is an independent prognostic factor of ccRCC and the establishment of a nomogram

Click here for additional data file.

Figure 6S TTC13 expression before and after mouse tumor xenografts experiments

Click here for additional data file.

We sincerely thank The Cancer Genome Atlas (TCGA) database for providing and managing patient data. The authors express their appreciation for the assistance by Dr Shaoqing Ju and Shanghai Changhai Hospital, Naval Medical University, which was instrumental in facilitating this research.

Additional Information and Declarations

Competing Interests

Author Contributions

Human Ethics

Data Availability

The authors declare there are no competing interests.

Lingling Xie performed the experiments, prepared figures and/or tables, authored or reviewed drafts of the article, and approved the final draft.

Yu Fang performed the experiments, prepared figures and/or tables, authored or reviewed drafts of the article, and approved the final draft.

Jianping Chen analyzed the data, authored or reviewed drafts of the article, and approved the final draft.

Wei Meng analyzed the data, prepared figures and/or tables, and approved the final draft.

Yangbo Guan conceived and designed the experiments, prepared figures and/or tables, and approved the final draft.

Wenliang Gong conceived and designed the experiments, performed the experiments, authored or reviewed drafts of the article, and approved the final draft.

The following information was supplied relating to ethical approvals (i.e., approving body and any reference numbers):

The study was approved and consented to by the Ethics Committee of the Affiliated Hospital of Nantong University(2022-K003-02) and the Naval Medical University, SYXK (Shanghai) 2022-0011. All patients provided written informed consent for the use of their tissue samples.

The following information was supplied regarding data availability:

The bioinformatics data (TTC13) is available at the TCGA (The Cancer Genome Atlas) database.

The hospital qPCR data is available in the Supplemental Files.

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
