# Peer review of "TTC13 expression and STAT3 activation may form a positive feedback loop to promote ccRCC progression"

_PeerJ, doi:10.7717/peerj.16316_

## Round 0.1 · original submission · Major Revisions

Please address the concerns of both reviewers and amend the manuscript accordingly.

Reviewer 1 ·

Basic reporting

The context and background provided to introduce the topics are appropriate. While the manuscript's content is overall understandable without significant issues, additional language editing is recommended to enhance readability. Below are some specific comments and suggestions on basic reporting:
- Line 5-7: "ccRCC, characterized by hematuria, pain, and the lump in the kidney area, accounts for about 75% of all kidney cancers; nevertheless, ccRCC is often asymptomatic or insidious at the early stage". This is confusing as these two statements appear contradictory. Please check and revise as necessary.
- In Figure 1, the current illustration for the 'Pan Cancer Analysis' could be improved for better representation. It's not immediately clear what the current image is intended to depict, so providing a more intuitive or descriptive image could enhance the figure's usefulness. Additionally, the image used for TTC13 appears to represent a nucleic acid secondary structure. TTC13 is a protein, not a nucleic acid, so this depiction is incorrect. To avoid potential confusion and enhance the scientific accuracy of your figures, please consider revising this depiction as well. Further, there also appears to be significant redundancy in content between Figure 1 and Figure 8. It is recommended to re-evaluate the necessity of both figures to determine if one can be removed or consolidated into the other.
- Figure 3A-B, 6B-C: please include what the error bars represent and the number of biological replicates performed in the figure captions.
- In the section 'TTC13-related signaling pathways in ccRCC' starting on line 147, it would be beneficial for the authors to provide more detail about how they selected the pathways for further investigation. For instance, the interferon-alpha, interferon-gamma, and E2F target pathways are all indicated as upregulated in Figure 5A, yet they do not appear to be discussed in the text.
- Please check and correct the labeling of the supplemental raw western blot data. The figure numbers labeled on the file names do not match the blot's positions in the Figures.
- Figures 6A, B, and C: it is unclear which inhibitors were used and at what concentrations. For clarity and to ensure reproducibility, it would be helpful if the authors could provide more detailed information.
- Line 170: "TTC13 activation" – the phrasing is unclear. Do the authors mean the supposed activation of the IL6-JAK-STAT3 signaling pathway by TTC13 or the overexpression of TTC13?
- The Discussion section of the manuscript contains a significant number of discussions of related studies. While it is appreciated and it is important to position your work within the broader scientific context, an excessive amount of discussion may be distracting. It is suggested to evaluate the relevance of the discussions and keep them concise and on topic.

Experimental design

There are some significant limitations to the experimental design and reporting of the experimental details. Below are some suggestions:
- For all methods used in this study, please ensure that full details of any commercial products used, including their catalog numbers, are provided. Also, it would be highly beneficial if the authors could include a more detailed stepwise procedure for each experiment. This information is critical for the reproducibility of the experiments.
- For better reproducibility and transparency, bioinformatics-related codes used should be included. Packages used should be cited, and relevant information on their version/build number should be provided.
- Line 43-46: antibodies are missing catalog numbers and dilutions.
- The methodology section indicates that a transient transfection was utilized. However, it is unclear whether this involved shRNA transfection or lentiviral transduction. This clarification is critical as it influences the interpretation and validity of long-term experiments, such as those involving cell line-derived xenografts. In general, transient transfections might not sustain the gene expression changes for the prolonged duration required in such experiments. Please clarify the methods used to ensure the robustness of the experimental findings.
- The pLTT3 plasmid was not introduced. Please provide the source of this plasmid.
- How the inhibitor of wnt/β-catenin or IL6-JAK-STAT3 was used in Figure 6 was not described.
- Were the blots stripped and reblotted, or are proteins run in separate western blots for figures containing multiple bands (i.e., Figure 5B-D, 6D-E)? Please indicate the procedures in the experimental methods.

Validity of the findings

While the bulk of the data presented appears to support the main hypothesis proposed by the authors, there are concerns about the validity of some findings. Specific comments and suggestions are below:
- It is recommended to add the blot of TTC13 in Figure 3D-E to validate the changes to their levels.
- Figure 4E presents IHC slides showing differences in protein intensities. However, image clarity is not sufficient for confidently assessing these differences. I would recommend supplementing the IHC data with a western blot. Furthermore, it would be helpful if the authors could provide more detail about the methods used to achieve TTC13 knockdown. If a transient knockdown system was employed, it is not immediately clear how the effects of the knockdown could be sustained for the 35-day period mentioned in the manuscript. Providing more information about the knockdown strategy and perhaps including additional data to confirm sustained knockdown over this timeframe would significantly strengthen this aspect of the study.
- Line 163-166: "To test this hypothesis, we performed rescued experiments using a specific inhibitor of wnt/β-catenin or IL6-JAK-STAT3 signaling pathway and found that the inhibitor could attenuate the growth-promoting effect of TTC13 individually, with synergistic effect when used in combination in CCK-8 and apoptosis assay (Figure 6A, B, C)" – the lack of detailed information about the specific inhibitors used and their respective concentrations hinders assessment of potential off-target effects. Moreover, the claim of a 'synergistic effect' should be supported by additional evidence, preferably via quantifiable metrics. Given that the inhibitors alone already induce a significant reduction in proliferation, it's not completely clear whether the effects observed with TTC13 are purely additive. To better elucidate this, I suggest performing a single timepoint dose-response (perhaps a 3-day treatment) using the wnt/β-catenin and IL6-JAK-STAT3 inhibitors, with NC or TTC13 modulation. By doing this, you could determine any change in IC50 and provide a stronger justification for any potentiation effect.
- For Figure 6D-E, similarly, the current presentation does not clearly differentiate between additive effects and potential potentiation or synergy. The western blots could benefit from including a blot for TTC13. As it stands, the interpretation of these results in this section is somewhat ambiguous. I suggest re-evaluating the claims made in this section to ensure they accurately reflect the data.
- In Figure 7E, it's not entirely clear from the provided images alone that p-STAT3 is indeed localizing to the nucleus. While some nuclear localization can be seen, the images may not sufficiently support this assertion without additional quantitative analysis. To provide better support, I recommend conducting a detailed quantification of the nuclear vs. cytoplasmic localization of p-STAT3. This can be achieved by analyzing a sufficient number of cells across multiple images and calculating the ratio of nuclear to cytoplasmic fluorescence for each cell. This would provide a more objective and robust measure of p-STAT3 localization.
- Lines 177-178 suggest that STAT3 might directly regulate TTC13 expression via a positive feedback loop mechanism to promote ccRCC cell proliferation. While this is a compelling hypothesis, it appears that more cellular-level validation is needed to conclusively establish this mechanism. I recommend that the authors perform additional validation experiments, including, but not limited to, (1) evaluating changes in endogenous TTC13 levels following treatment with AG490; (2) investigating changes in endogenous TTC13 levels following STAT3 knockdown or overexpression; and/or (3) assessing whether the STAT3 modulated changes in TTC13 correspond to expected changes in ccRCC cell proliferation.

Additional comments

This is an overall very interesting manuscript that implicates TTC13 in ccRCC progression. I commend the authors for their efforts. However, there are some significant limitations in some aspects of the manuscript that may need to be addressed. Specific comments are provided in their respective sections.

·

Basic reporting

The manuscript exhibits a lack of professional writing. Numerous grammatical and formatting errors are present throughout, which could significantly affect its overall quality. I highly recommend that the author consider employing the services of a professional editing service to rectify these errors and enhance the overall presentation of the manuscript.

Experimental design

This study is meticulously designed.

Validity of the findings

The results presented in this study effectively and comprehensively address the core research question, providing valuable insights and contributing significantly to the field.

Additional comments

Minor concerns:
1. Figure 1 should be deleted or moved to the last position.
2. The qualification of WB in Figure 2E should included.
3. In Figure 3, the TTC13 low cell line HK-2 should included as a control.
4. Qualification should be added to Figure 3C.
5. In Figure 4E, IHC qualification should be included. Additionally, the reason why knockdown TTC13 and Stat3 show a more nuclear form compared to the control should be addressed. Furthermore, cleaved caspase 3 staining should be included.
6.WB qualification should included in Figure 5.
7. The mechanism of how TTC13 regulates Stat3 activity should included. TTC13 directly phosphorylates STAT3? The direct relationship is not clearly defined or apparent.
8. Qualification should be added to Figure 6A.
9. Autophagy has been implicated in promoting cancer cell survival. Considering this, it would be beneficial to explore the combined treatment of an autophagy inhibitor and a JAK inhibitor in ccRCC.
10. In Figure 7E, the labeling for "p-STAT3" should be changed to "STAT3", since p-STAT3 is expected to be in the nuclear form consistently.

---

## Round 0.2 · Minor Revisions

As you can see, the reviewer was mostly satisfied by your response and revsion. However, a few minor issues were pointed out. Please address these remaining issues pointed by the reviewer and amend manuscript accordingly.

Reviewer 1 ·

Basic reporting

The manuscript has improved and the authors have addressed most of the concerns from the first round of review. Below are a few additional minor comments:
- There appear to be some missing spaces in lines 230-231 – i.e., “TTC13overexpressionincreased”.
- Figure 5 caption: “AG490 concentration: 50µmol; ICG001:20µmol” – µmol itself is not a unit for concentration. I assume it is µmol/L or µM? Please check and revise concentration reportings where they appear in the manuscript.
- Figure 6D, correct the spelling of “nucleus”.
- The bar graph of the quantification of Figure 3F is not appropriate. It is difficult to compare the increases and decreases.
- Lentiviral transduction methods should be included or cited.

Experimental design

no comment

Validity of the findings

no comment

---

## Round 0.3 · accepted · Accept

All remaining concerns of the reviewer were adequately addressed and the manuscript was revised accordingly. Therefore, the amended version is acceptable now.